# Identification of MicroRNAs as Viable Aggressiveness Biomarkers for Prostate Cancer

**DOI:** 10.3390/biomedicines9060646

**Published:** 2021-06-05

**Authors:** Luis Javier Martínez-González, Victor Sánchez-Conde, Jose María González-Cabezuelo, Alba Antunez-Rodríguez, Eduardo Andrés-León, Inmaculada Robles-Fernandez, Jose Antonio Lorente, Fernando Vázquez-Alonso, María Jesus Alvarez-Cubero

**Affiliations:** 1GENYO. Centre for Genomics and Oncological Research: Pfizer, University of Granada, Andalusian Regional Government, Genomics Unit, PTS Granada-Avenida de la Ilustración, 114-18016 Granada, Spain; alba.antunez@genyo.es; 2Urology Department, Hospital Virgen de las Nieves, 18014 Granada, Spain; vaitor1991@gmail.com (V.S.-C.); fvazquezalonso@gmail.com (F.V.-A.); 3Research and Development Department, Meridiem Seeds, 04710 Almería, Spain; jm.gonzalez.cabezuelo@gmail.com; 4Bioinformatics Unit, Institute of Parasitology and Biomedicine “López-Neyra” (IPBLN), Spanish National Research Council (CSIC), 18016 Granada, Spain; eduardo.andres@csic.es; 5GENYO. Centre for Genomics and Oncological Research: Pfizer, University of Granada, Andalusian Regional Government, Liquid Biopsy and Cancer Interception Group, PTS Granada, 114-18016 Granada, Spain; inmaculada.robles@genyo.es (I.R.-F.); jose.lorente@genyo.es (J.A.L.); 6University of Granada, Legal Medicine and Toxicology Department, Faculty of Medicine, PTS Granada, 18016 Granada, Spain; 7University of Granada, Department of Biochemistry and Molecular Biology III, Faculty of Medicine, PTS Granada, 18016 Granada, Spain; 8Nutrition, Diet and Risk Assessment Group, Bio-Health Research Institute (ibs.GRANADA Instituto de Investigación Biosanitaria), 18014 Granada, Spain

**Keywords:** aggressiveness, biomarkers, bioinformatic, precision medicine, prostate cancer

## Abstract

MiRNAs play a relevant role in PC (prostate cancer) by the regulation in the expression of several pathways’ AR (androgen receptor), cellular cycle, apoptosis, MET (mesenchymal epithelium transition), or metastasis. Here, we report the role of several miRNAs’ expression patterns, such as miR-93-5p, miR-23c, miR-210-3p, miR-221-3p, miR-592, miR-141, miR-375, and miR-130b, with relevance in processes like cell proliferation and MET. Using Trizol^®^ extraction protocol and TaqMan™ specific probes for amplification, we performed miRNAs’ analysis of 159 PC fresh tissues and 60 plasmas from peripheral blood samples. We had clinical data from all samples including PSA, Gleason, TNM, and D’Amico risk. Moreover, a bioinformatic analysis in TCGA (The Cancer Genome Atlas) was included to analyze the effect of the most relevant miRNAs according to aggressiveness in an extensive cohort (*n* = 531). We found that miR-210-3p, miR-23c, miR-592, and miR-93-5p are the most suitable biomarkers for PC aggressiveness and diagnosis, respectively. In fact, according with our results, miR-93-5p seems the most promising non-invasive biomarker for PC. To sum up, miR-210-3p, miR-23c, miR-592, and miR-93-5p miRNAs are suggested to be potential biomarkers for PC risk stratification that could be included in non-invasive strategies such as liquid biopsy in precision medicine for PC management.

## 1. Introduction

Current medicine is focused on precision medicine, with the aim of providing better management and treatment of patients in several diseases like cancer. One of the main aspects of precision medicine is to provide optimal biomarkers, to offer more accurate strategies for disease detection, diagnosis, prognosis, prediction of response to intervention, and disease monitoring.

PC is one of the most prevalent (incidence of 37.5 per 100,000) tumors among men in the world after lung cancer and the second most frequent cause of cancer-related death (375,000 deaths worldwide) [1]. It also accounts for around the 20% of newly diagnosed cancers, according to 2020 cancer statistics [2]. However, there are not many biomarkers used into clinical practice. Liquid biopsy in cancer has gained momentum in clinical research and it is experiencing a boom for various applications. For instance, exosome-based biomarkers have quickly become adopted in the clinical arena and the first PC exosome RNA-based test has already helped >50,000 patients in decision processes. As a result, it is now included in the National Comprehensive Cancer Network (NCCN) guidelines for early PC detection [3]. Moreover, recent studies have evaluated the role of a circular RNA, known as circANKS1B, as a potential prognostic biomarker and therapeutic target for PC. This circANKS1B has been suggested to act as a sponge for miR-152-3p, promoting PC progression by upregulating TGF-α expression [4]. Others include the role of miR-144 as a potential biomarker for predicting PC progression, by its interaction with EZH2, reducing cell viability and promoting cell apoptosis [5], or the role described for miR-145-5p in PC metastasis by binding *TOP2A*, which is indicated as a useful biomarker for metastatic PC detection [2]. However, there are also data including miR-320a as a valuable biomarker that can be used in early diagnosis of PC [6].

We focused, in the present work, on several miRNAs based on carcinogenesis or tumor aggressiveness. In this regards, miR-93-5p, in combination with other miRNAs (miR-17-5p, miR-20a-5p, miR-92a-3p), has been appointed as a potential signature in PC, by the regulation of relevant genes such as *E2F2*, *RRM2*, and *PKMYT1* [7]. Additionally, miR-23c has been recently indicated as a biomarker for predicting recurrence in PC, by its critical role on *RGN* (Regucalcin) gene. RGN promotes dormancy in PC [8]. Moreover, miR-23c has been described as an inhibitor of cell proliferation and a promoter of apoptosis by attenuating *ERBB2IP* (erbb2 interacting protein) [9].

In the case of miR-210-3p, it was previously included as a potentially non-invasive biomarker related to diagnosis and treatment management in clear cell renal cell carcinoma [10]. In PC, several results indicated that an up-regulation of miR-210-3p activates *NF-κB* signaling pathway, which is related to bone metastasis in PC, or even targets *FGFRL1*, which promotes lung cancer metastasis [11,12]. Moreover, there are also reports about miR-210-3p-EphrinA3-PI3K/AKT signaling axis and its role in oral squamous cell carcinoma progress and treatment [13]. Recent data exposed by Ruiz-Plazas et al. reinforced the relevant role of miRNAs as biomarkers in PC. Ruiz-Plazas et al. discovered, in 97 semen samples from PC patients, the efficiency of exo-oncomiR-221-3p, miR-222-3p, and *TWEAK* as biomarkers for classification of aggressive PC patients (with 85.7% specificity and 76.9% sensitivity) [14]. However, miR-221-3p is also related to breast cancer as a new strategy for improving success in chemotherapy [15].

Deregulation of miR-592 was previously reported in other tumors such as high-grade uveal melanoma, renal cell carcinoma, colorectal carcinoma with unaffected mismatch repair mechanisms, and gastric cancer. Direct interactions with genes involved in cell migration, motility, development, and regulation of cell signaling have also been deciphered [16]. As can be seen, there are many aspects that indicate miR-592 with interest for cancer aggressiveness biomarker and also for prognosis and therapeutics in patients with renal cell carcinoma. An over-expression of miR-592 is suggested to promote proliferation, migration, and invasion of renal cell carcinoma cells by targeting *SPRY2* (gene related to *PI3K*/*AKT* and *MAPK*/*EPK* signaling pathways) [17]. In PC, there are also data relating miR-592 with aggressiveness in combination with *ANPEP*, miR-217, and miR-6715b [18]. miR-375 has been included as one of the classical miRNAs’ biomarkers in PC. It has importance in several points of the disease, such as clinical T-stage and bone metastasis, distinguishing between PC patients versus controls, and even differentiating between benign prostatic hyperplasia (BPH) in urinary and serum exosomes [19,20,21]. This miRNA has been included as one of the main points for regulating proliferation, metastasis, and EMT in PC, and it has also been included with a significant down-regulation after radical prostatectomy in urine extracellular vesicles (EVs), clarified urine, and blood plasma [22].

One of the first proposals for using circulating miRNAs for PC screening was focused on the ratios of several miRNAs, such as miR-106a/miR-130b and miR-106a/miR-223, ratios for discriminating localized PC versus BPH patients [23]. However, there are more reasons suggesting the relevant role of miR-130b in PC, such as miR-130b/miR-301b with clinical variables reporting positive correlations with malignancy, T-stage, residual tumor status, and primary therapy outcome [24]. The role of this miRNA and tumorigenesis is mainly described by its effect on the down-regulation of *MMP2* (matrix metalloproteinase-2), which has a suppressive effect on PC metastasis [25]. Moreover, recent data have attributed a function as a therapeutic target in anti-angiogenesis treatment by the correlation of miR-130b/*TNF-α*/*NF-κB*/*VEGFA* feedback loop with angiogenesis in PC [26].

MiR-141 has been suggested as one of the most interesting markers for PC diagnosis and even treatment [27]. Recent publications discovered that exosomal miR-141-5p levels showed a slight increase in PC patients, so it is included as a useful specific marker for PC [28] or even to distinguish among PC and BPH [29].

However, it is also important to take into account the role of germline, somatic, or epigenetic mutations as new, promising, molecular networks and signaling pathways implicated in aggressive PC, such as *STAT3*, *PTEN*, *ATM*, *AR*, and *P53* [30], or cfDNA (cell free DNA) and exosome-RNA, as reliable sources of AR variants, and their combined detection in liquid biopsy that predicts resistance to *AR* signaling inhibitors in PC [31].

In conclusion, the election of an effective biomarker for PC is controversial. Recent computational analysis indicated that miRNAs biomarkers along with the associated miRNA–mRNA relationships are good options for PC management and carcinogenic deciphering [32]. For that reason, we focused this study, including a complete analysis combining bioinformatics’ strategies and laboratory assays, to prove the role of miR-93-5p, miR-23c, miR-210-3p, miR-221-3p, miR-592, miR-141, miR-375, and miR-130b in PC aggressiveness and diagnosis.

## 2. Materials and Methods

### 2.1. Study Subjects

Samples from fresh tissue of 131 men with confirmed PC and 28 fresh tissue samples of control patients were included in the present study. Subjects with prostate-specific antigen (PSA) levels ≥4.0 ng/mL, meeting the criteria for undergoing a prostate biopsy, were recruited from 2012 to 2014 and included in this study (Table 1). Samples classified as controls were those with negative biopsy values. All individuals underwent a systematic, 20-core ultrasound-guided biopsy in order to limit the false-negative rates. Those with positive biopsy for PC were analyzed for T stage, serum PSA levels, and Gleason score, and then classified according to D’Amico risk classification (low, intermediate, and high risk). Clinical follow-up for 60 months was performed by urologists from the “Hospital Virgen de las Nieves, Granada, Spain”. Moreover, 60 plasmas from peripheral blood collected in EDTA tubes, from the same men of the previously collected tissue samples, were also included in the analysis for proving liquid biopsy biomarkers. Samples were collected and stored at −80 °C until processing. All participants provided a written, informed consent, and the study was previously approved by the Research Ethics Committee of Granada Centre (CEI-Granada internal code 1638-N-18) following the Helsinki ethical declaration.

### 2.2. Bioinformatic Analysis

The Cancer Genome Atlas (TCGA) was launched as one of the main projects accelerating the comprehensive understanding of the genetics of cancer using innovative genome analysis technologies, helping to generate new cancer therapies, diagnostic methods, and preventive strategies [33,34]. The structure of TCGA is well organized and involves several cooperating centers responsible for collection and sample processing, followed by high-throughput sequencing and sophisticated bioinformatics data analyses [35]. This makes TCGA repositories extraordinary-value sources of data in studies of the characteristics of the present work. By using prostate adenocarcinoma (PRAD) repository information from TCGA Program, a miRNAs’ differential expression analysis was carried out to identify differentially expressed miRNAs that could potentially serve as biomarkers.

#### 2.2.1. TCGA Data of Prostate Adenocarcinoma (TCGA_PRAD)

A total of 531 isoforms’ expression quantification (miRNA-Seq) files containing tumor (T; *n* = 480) and non-tumor (NT; *n* = 51) tissue samples, as well as clinical data for each sample, were obtained from the TCGA data portal (https://portal.gdc.cancer.gov, accessed on 04 June 2021). Out of the 480 T samples, 285 were from patients with Gleason score ≤7 (G1) and 195 corresponded to patients with Gleason score >7 (G2). The NT samples were included in the G0 group.

#### 2.2.2. Differential Expression (DE) Analysis

TCGA_PRAD differential expression analyses were carried out using edgeR (version 3.28.0) Bioconductor package [36,37]. The GLM (Generalized Linear Models) approach was used to include an experimental design with Gleason score as factor. Quasi-likelihood F-test (QL) was used to determine differentially expressed (DE) miRNAs related to tumor aggressiveness. Three categories were defined: (1) G0 = NT samples (51 cases), (2) G1 = Gleason score ≤ 7 (285 cases), and (3) G2 = Gleason score > 7 (195 cases). Lowly expressed miRNAs (miRNAs that did not reach 1 count per million in a minimum of 51 samples) were filtered out. By using this filter, 1564 miRNAs were removed from the initial 2095. Samples were normalized by using the trimmed mean of M-values (TMM) method [38]. Three contrasts were carried out in the analysis: G0 vs. G1, G0 vs. G2, and G1 vs. G2. The *p*-value was adjusted by Benjamini–Hochberg false discovery rate (FDR) procedure [39]. Differentially expressed miRNAs were those having a fold change |FC ≥ 1.5| and a FDR < 0.05. Principal Component Analysis (PCA) algorithm [40] was used for graphic representation of samples distribution based on DE miRNA by using the M3C (version 1.8.0), ggrepel (version 0.8.1), and ggplot2 (version 3.0.1) packages in the R environment.

### 2.3. Molecular Analysis

Total RNA of 159 fresh tissue biopsies were extracted using Trizol^®^/chloroform method and quality validated by A260/A280 in NanoDrop™ 2000c (Thermo Fisher Scientific, Inc., Wilmington, DE, USA). Reverse transcription was performed with TaqMan™ Advanced miRNA cDNA Synthesis kit (Applied Biosystem, Foster City, CA, USA). Quantitative polymerase chain reaction (qPCR) was performed with TaqMan probes (Life Technologies, Carlsbad, CA, USA), according to the manufacturer’s protocol, on a 96-wells plate with QuantStudio 6 Flex Real-Time PCR System (Applied Biosystems). The qPCR reactions were performed as follows: 5 °C during 20 s for enzyme activation, followed by 40 cycles of 1 s at 95 °C and 20 s at 60 °C for denaturing and annealing/extension. For the liquid biopsy analysis, plasmas of 60 samples were isolated from blood. This process was carried out, at most, 4 h after collection. Total RNA of the samples was extracted using the miRNeasy Serum/Plasma Kit (GE; Qiagen, Hilden, Germany). All samples were run in triplicates, with a NTC (non-template control) in each plate; Ct_s_ ≥ 35 was considered as undetermined value.

### 2.4. Selection of Candidate miRNA Normalizers

First, we chose among previously published data the most effective housekeeping, selecting GAPDH, RNU6B, and miR-130b as possible effective candidates. We analyzed those displaying the highest stability and the lowest biological variance in PC. After developing an assay of qPCR, as explained above, we analyzed the mean, standard deviation (SD), and variation coefficient (CV) comparing with Ct (see details in Appendix A). Moreover, to choose the best housekeeping we used several computational programs such as NormFinder [41] (https://moma.dk/normfinder-software) and BestKeeper [42] (http://www.gene-quantification.de/bestkeeper.html, accessed on 04 June 2021) to assure the stability of endogenous housekeeping in PC (details in Appendix A). Finally, RNU6B (Ct mean = 29.446) was chosen by its higher stability in these samples (see details in Appendix A).

### 2.5. Selection of the miRNAs

After combining bioinformatic analysis with the most appropriate miRNAs, according to pubMed publications (including biomarkers, cancer aggressiveness, and PC), we finally chose eight miRNAs for the experimental analysis in tissues and plasmas from peripheral blood samples. From the bioinformatics reports, we selected DE miRNAs (see details in Appendix A), and all those miRNAs in top position and with confirmed role in cancer or PC development were selected. Therefore, miR-592, miR-23c, miR-93-5p, miR-210-3p, miR-141, miR-375, miR-130b, and miR-221-3p (see Appendix A) were included in the present analysis. miRNAs’ expression levels were quantified using the comparative threshold cycle (Ct) method (2^−ΔΔCt^), relative to RNU6B expression as an endogenous control.

### 2.6. Statistical Analysis

SPSS v.22 (IBM Inc, Armonk, NY, USA) and GraphPad Prism v.5 software (GraphPad Software Inc, La Jolla, CA, USA) were used for analysis and graphic designing. A *p* value < 0.05 was considered as significant value. miRNAs expression patterns’ comparisons were done using value 2^−Δ∆CT^. Kolmogorov–Smirnov test was developed to assess normal distribution of all data. For determining significant differences among miRNAs’ expression levels and clinical variables, a non-parametric U Mann-Whitney test was performed.

## 3. Results

### 3.1. Bioinformatic Analysis

A bioinformatic analysis using TCGA repository was developed to select the most appropriate miRNAs in PC.

#### 3.1.1. TCGA Data of Prostate Adenocarcinoma (TCGA_PRAD)

First, TCGA analysis reported several top miRNAs as up-regulated and down-regulated with effect in PC aggressiveness according to Gleason Score. We developed two main comparisons, selecting samples with reported paired results reported in a healthy area (G0) of PC tissue and an affected tumor area, with Gleason scores equal to or below 7 (G1) and above 7 (G2) (see details in Table 2), G0 vs. G1 and G0 vs. G2 (in Appendix A).

Sampl−s were analyzed with PCA, and a clear difference among non-tumoral (G0) versus tumoral (G1 and G2) was denoted. However, not many dispersions of the data were remarked among Gleason score equal to or below 7 (G1) compared to those above 7 (G2). As can be seen, the majority G1 samples were closer to G0 (more details in Figure 1).

#### 3.1.2. Differential Expression (DE) Analyses

When performing DE analysis, we found that 9.7% of all miRNAs were in common when comparing non-tumoral vs. tumoral samples. Therefore, these miRNAs could be common pathways in prostate. The highest differences were found when comparing non-tumoral (G0) versus tumoral samples with Gleason scores above 7 (G2) (more details in Figure 2).

Volcano plots (Figure 3) represent miRNAs dispersion according to their expression patterns, using logFC (X axis) versus statistical significance FDR (Y axis). Blue represents those miRNAs that are not differentially expressed contrasting with DE analysis, which is represented in red. Nearer to the left side are miRNAs under-expressed contrasting with those closer to the right side, which are over-expressed. Moreover, those miRNAs that are on the upper part of the Y axis had a stronger statistical significance. The most interesting miRNAs for the present work are identified as surrounded by a circle in Figure 3. We have also represented, using volcano plots, both comparisons, G0 versus G1 and G0 versus G2 (see details in Appendix A).

### 3.2. Molecular Analysis

Those miRNAs with stronger evidence in PC, according to publications and previously described bioinformatic analyses, were then validated in a cohort of 159 fresh tissue samples (see details in Table 1) and 60 plasma samples. Although some of these miRNAs, such as miR-141, miR-375, and miR-130b, did not have relevant positions according to our bioinformatic analysis, we just included them for having over-expressed values with low log-FDR (see Table 2, Appendix A, Figure 3, Appendix A). By contrast, these are some of the most significant miRNAs according to previous publications [19,24,43,44,45]. Therefore, we decided to include them in the validation process by doing laboratory experiments. However, we could not find any significant statistical value when comparing with clinical parameters such as PSA, Gleason score, T stage, and D’Amico risk (more details in Appendix A).

As can be seen in Figure 4, the expression trend in both studies, with our cohort and TCGA data, is the same for all miRNAs, except for miR-212-3p and miR-23c. According to the remaining miRNAs selected based on TCGA analysis, we developed two main clusters.

One included miR-210-3p, miR-23c, and miR-592, which had concordance in experimental and bioinformatic analysis. All these three miRNAs demonstrated an interesting value as PC aggressiveness biomarkers (G1 vs. G2 comparisons). When comparing bioinformatic data (TCGA) in our analysis with those data from qPCR, miR-210-3p showed a high expression pattern in G2 vs. G1 (*p* = 1.89 × 10^−23^). This pattern was also maintained in qPCR data in biopsy samples (*p* = 0.001). In the case of miR-23c and miR-592, the same expression patterns were also followed in G1 and G2, but with no statistically significant values (details in Appendix A). Surprisingly, when comparing G0 vs. G1 and G0 vs. G2 we found in miR-23c a great change in TCGA analysis versus qPCR in tissue samples. Moreover, miR-592 could be a follow-up biomarker because it had the same tendency in TCGA and in qPCR analysis, but its low expression made it difficult to obtain plasmas from peripheral blood characterization.

Secondly, miR-93-5p showed the same patterns in bioinformatic (TCGA) and experimental analysis (qPCR) in all comparisons (G1 vs. G2) (see details in Figure 4 and Appendix A). Moreover, miR-93-5p and miR-210-3p were the ones with the most statistically significant as diagnostic biomarkers (more details in Appendix A).

Moreover, when trying to find the most suitable biomarker for liquid biopsy, we believed that miR-93-5p was the best choice. As we have previously mentioned, it followed the same patterns in all analyses (TCGA and qPCR) and in all samples (plasmas from peripheral blood and tissues), and it had statistically significant values in most of the analysis (Appendix A) (more details in Appendix A).

### 3.3. Predicted Functional Analysis

In order to identify potential target genes for miRNAs associated with aggressiveness and diagnosis (miR-210-3p, miR-23c, miR-592, and miR-93-5p), we performed an integrated target prediction using two different databases: Mirtarbase (http://mirtarbase.cuhk.edu.cn, accessed on 04 June 2021) and MiRWalk3.0 (http://mirwalk.umm.uni-heidelberg.de, accessed on 04 June 2021). The miRNA target genes indicated by both prediction programs, especially those related to carcinogenic processes, were identified.

In a subsequent analysis using IPA (Ingenuity Pathway Analysis) [46], STRING (https://string-db.org/) [47], and DAVID Bioinformatics Resources v6.8 (https://david.ncifcrf.gov, accessed on 04 June 2021) [48], we obtained the role of gene set, clinical implication, ontology, and involved metabolic pathways. As a result of this approach, we were able to identify two target genes for miR-93-5p (i.e., STAT3 and IGF2) that may be involved in the development and progression of PC (Figure 5). Thus, we focused the functional analysis on both genes to see pathways’ interactions (more details in Appendix A).

## 4. Discussion

Looking for efficient biomarkers for PC diagnosis and stratification is one of the most important gaps between research and medicine in managing PC. Diagnostic, monitoring, and prognosis biomarkers in PC are still well undefined. There is much research focusing on looking for a stable molecule that could help in PC management. miRNAs profiling has shown important results, such as the use of miRNAs present in semen exosomes (miR-142-3p, miR-142-5p, and miR-223-3p) as models based on molecular biomarkers for improving PC diagnosis/prognosis efficiency [49] as well as the use of miR-17, miR-20a, miR-20b, and miR-106a signatures in blood that can distinguish high- and low-risk PC patients after radical prostatectomy [50] or urinary exosomes’ microRNAs (miR-196a-5p and miR-501-3p) as non-invasive biomarkers [51]. The stability of miRNAs, combined with the increased range of methodologies for their study and their key role in several regulatory pathways such as *EMT*, angiogenesis, metastasis, and drug resistance [52], makes miRNAs one of the most promising options as biomarkers in PC. It is true that most of the research is focused on miRNAs. Nevertheless, they are not the unique, non-invasive biomarkers being currently studied. Others, such as neutrophil-to-lymphocyte ratio [53], *DLX1*, *PCA3*, and *DUOX1* mRNA plasma expressions [54] or a 14-genes’ panel urine test including *PMP22*, *GOLM1*, *LMTK2*, *EZH2*, *GSTP1*, *PCA3*, *VEGFA*, *CST3*, *PTEN*, *PIP5K1A*, *CDK1*, *TMPRSS2*, *ANXA3*, and *CCND1* [55], are some of the most updated research in PC non-invasive biomarkers. However, as previously mentioned by several authors, such as H.P. Liu et al., miRNAs are suitable molecules for being biomarkers in PC with sensitivity, specificity, and accuracy values next to 99% in extended cohorts [56].

It is true that liquid biopsy is one of the main focuses of attention in current research, with the aim to apply it to the biomedical field. It is known that, compared to tissue samples, liquid biopsies are of particular interest in clinical settings due to their minimal invasiveness since they provide the opportunity to repeat sampling, as well as to have a whole representation of the entire tumor [57]. Liquid biopsies could be a relevant point when there is limited access to the tissue biopsy, giving increasing advantages in precision medicine for direct biomarker application in clinical practice [58]. Moreover, PC is a heterogeneous tumor and there are discrepancies in published data. Therefore, a deep analysis combining bioinformatics and molecular experimental analysis could be the clue to improve the use of miRNAs as biomarkers in PC.

Present bioinformatics analysis gave us results of major miRNAs, such as miR-891a-5p, miR-23c, miR-93-5p, miR-145-3p, and miR-221-3p, when comparing NT versus T values with Gleason score above 7, or miR-592, miR-508-3p, miR-210-3p, miR-514a-3p, and miR-509-3p, when comparing G1 vs. G2 in tumoral samples. However, not all of them have been pointed out with a relevant role in PC or tumorigenesis effect, such as miR-891a-5p, miR-145-3p, miR-508-3p, miR-514a-3p, or miR-509-3p, which were discarded in subsequent molecular analysis. The remaining miRNAs were selected with a strong relation in publications and bioinformatics analysis in PC or tumor aggressiveness. However, several miRNAs, such as miR-141, miR-375, and miR-130b, lost interest as biomarkers when performing expression analysis in the present study because there were no differences among PC risk or even prognosis. Nevertheless, when looking in literature, these are among the most highlighted miRNAs as suitable biomarkers in urine, for PC diagnosis and progression [19,21]; in plasma, for time to progression of metastatic castration-resistant PC or biomarker screening [20,23,24,59]; or even associated with an increased risk of biochemical recurrence [60]. By contrast, when studying them in the present bioinformatics analyses, these three miRNAs (miR-141, miR-375, and miR-130b) completely lost total statistical power (Figure 3 and Figure 4).

When comparing the remaining potential miRNAs, according to the present bioinformatics analysis (miR-93-5p, miR-23c, miR-210-3p, miR-221-3p, and miR-592), all were suggested with important roles in PC screening [61], aggressiveness [8,62,63], and recurrence [7,18,64], or even new etiological values [65] were suggested. These miRNAs have also been highlighted with carcinogenic effects in other tumors, such as endometrial [66,67], osteosarcoma [68], hepatocellular [69], renal [17,70], or colorectal [71], among others [72,73,74]. According to our experimental analysis, miR-210-3p, miR-23c, and miR-592 were suggested as aggressiveness biomarkers in PC and miR-93-5p, miR-130b, and miR-141 were suggested as diagnostic biomarkers for this tumor.

Moreover, if we try to develop non-invasive biomarkers, we will select miR-93-5p for diagnostic purposes, according to the obtained data in TCGA, and analysis in tissue and plasmas from peripheral blood samples. miR-210-3p and miR-592 should also be good options. Even though the obtained results in plasmas from peripheral blood samples of the present study were scarce, TCGA and tissue analysis showed relevant data. Although we know that one of the limitations of the present study is the small amount of plasma from peripheral blood samples included in these experiments, it is certain that the same tendency was shown in all of the present samples, which allows us to encourage the role of non-invasive biomarkers and to face future analysis in liquid body samples. Moreover, previously published data indicated the promising role of miR-93-5p for predicting the disease aggressiveness with diagnosis accuracy in liquid biopsy [75] or even associated with lymphatic dissemination [66]. Here, we reinforced the role of miR-93-5p as a biomarker in PC because all present analyses (bioinformatics and experimental) showed the same patterns. Moreover, it was related to *IGF2* and *STAT3* genes, according to the integrated target prediction. Both genes have proven oncogenic effects, such as the role of acetylated STAT3-mediated activation of *IGF2* transcription in *HDI* (Histone deacetylase inhibitors) resistance described in NSCLC (non-small-cell lung carcinoma) [76]. There are also some reports in PC. Furthermore, the role of IGF2 messenger RNA binding protein 3 (*IMP3*) produces an acceleration in PC progression through activating *PI3K*/*AKT*/*mTOR* signaling pathway via increasing SMURF1-mediated *PTEN* ubiquitination [77]. All these data reinforce our results about the promising use of miR-93-5p as PC biomarkers. It was previously only suggested by bioinformatics analysis in the study of Y. Yang et al. [78]. Moreover, miR-93-5p in hepatocellular carcinoma has been proven to bind to the 3′-untranslated region (UTR) of mitogen-activated protein kinase kinase kinase 2 (*MAP3K2*). *MAP3K2* directly up-regulated its expression and down-regulated p38 and c-Jun N-terminal kinase (JNK) pathway. All these pathway connections lead to cell cycle progression in hepatocellular carcinoma and also explain its possible role in PC [79]. Moreover, it has been described that *AHNAK* is the target gene of miR-93-5p. This gene is a large, scaffolding protein, which has also been identified as acting as a tumor suppressor and which is highly related to tumor metastasis. It has been suggested that *AHNAK* plays an inhibitor role in migration and invasion, as well as *EMT* in cancer [80]. Therefore, we hypothesized that miR-93-5p may also be involved in migration and invasion or *EMT* process in PC, as it has been described in gastric, hepatocellular, and NSCLC cancer. It was previously suggested by Yang K et al., who indicated that miR-93 would function as a tumor promoter in PC by targeting disabled homolog 2, and Liu JJ et al., who found that miR-93 could promote the proliferation and invasion of PC cells by upregulating their target genes *TGFBR2*, *ITGB8*, and *LATS2* [81].

By contrast, there is just one publication that indicated the role of miR-23c in PC, suggesting its role in dormancy [8]. Other reports suggest this miRNA has a role as a therapeutic target in ovarian cancer [68] and implicate the molecular regulation of endometrial or hepatocellular cancers [66,69]. According to KEGG Pathways, miR-23 up-regulates *MAPK1* and *FGFR3* genes in bladder cancer. Both these genes have been seen in many cancers at different stages, such as proliferation, by interactions with *RAS*-*MAPK* pathways. These genes are confirmed for having an oncogenic potential role across multiple cancer types; indeed, *MAPK1* is altered in 0.82% of all cancers [82]. Although in PC there are scarce data about the role of miR-23c, it makes sense what we have proven in the present article concerning its role as biomarker for aggressiveness. According to the Atlas of Genetics and Cytogenetics in Oncology and Haematology, in prostate tumors the level of activated MAP kinase was increased with higher Gleason score and tumor stage, while non-neoplastic prostate tissue showed little or no staining with activated MAP kinase antiserum, which makes sense with the miR-23 c correlation in PC in the present report [83].

The remaining suggested biomarkers in the present study, such as miR-210-3p, have been previously reported in relation to activating the NF-κB signaling pathway in PC [11] and miR-592 distinguishing between T2c to T3b PC stages [18]. The present data, although they are scarce, are in the same line with previous reports, linking these miRNAs with PC aggressive stages.

The novelty of the present work is that it gives strong associations using three different strategies (bioinformatics and expression analysis in tissue samples and in plasma samples), providing a correlation between these analysis in miRNAs and PC.

## 5. Conclusions

In summary, our study is one of the first reports combining bioinformatics and experimental analysis (in plasma and tissue PC samples) for validating miR-210-3p, miR-23c, miR-93-5p, and miR-592 as promising biomarkers in PC. According to our results, we will suggest that miR-93-5p is the most promising non-invasive biomarker for PC aggressiveness and diagnosis. This is the first time that miR-23c has been related to PC. Future analysis in these miRNAs, using more samples of plasmas from peripheral blood and urine, could be one of the most important steps for finally including these miRNAs as non-invasive biomarkers for precision medicine in PC.

## Figures and Tables

**Figure 1 biomedicines-09-00646-f001:**
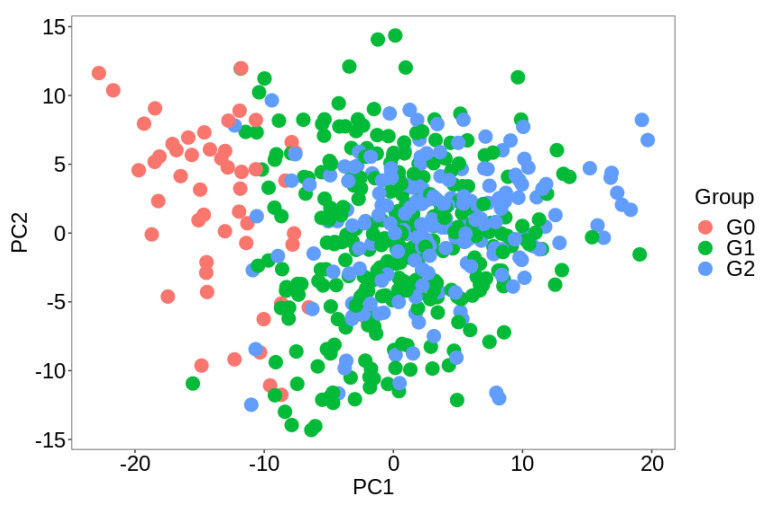
PCA plot representing G0, G1, and G2 TCGA data.

**Figure 2 biomedicines-09-00646-f002:**
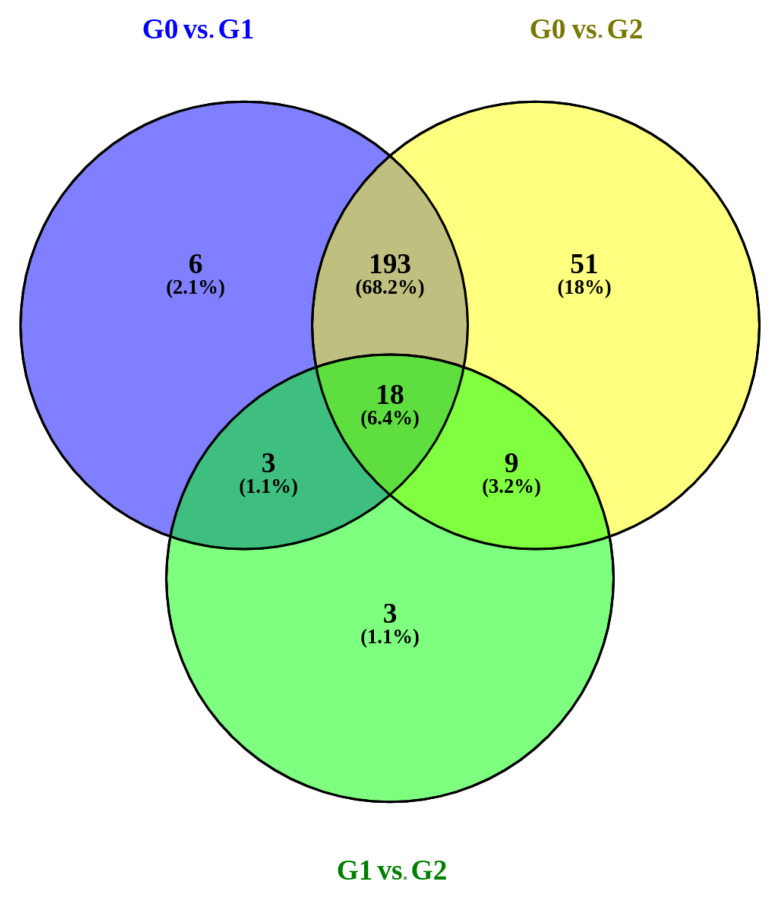
Venn diagram showing DE miRNA in the three differential expression analyses performed in TCGA. Intersections represent those miRNAs commonly differentially expressed.

**Figure 3 biomedicines-09-00646-f003:**
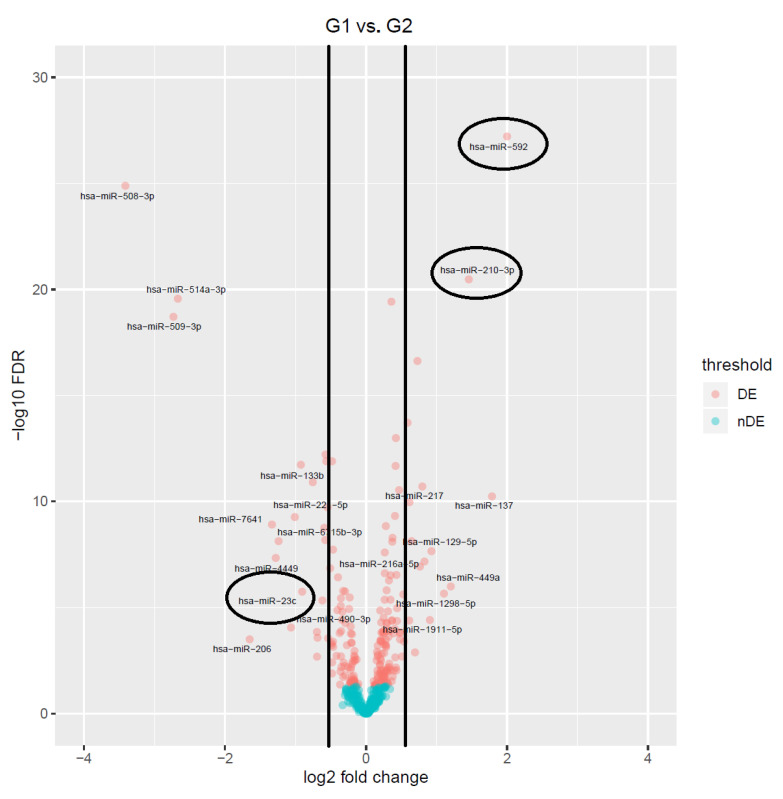
Volcano Plot comparing tumoral samples with Gleason score ≤ 7 (G1) with tumoral samples with Gleason score > 7 (G2) in TCGA.

**Figure 4 biomedicines-09-00646-f004:**
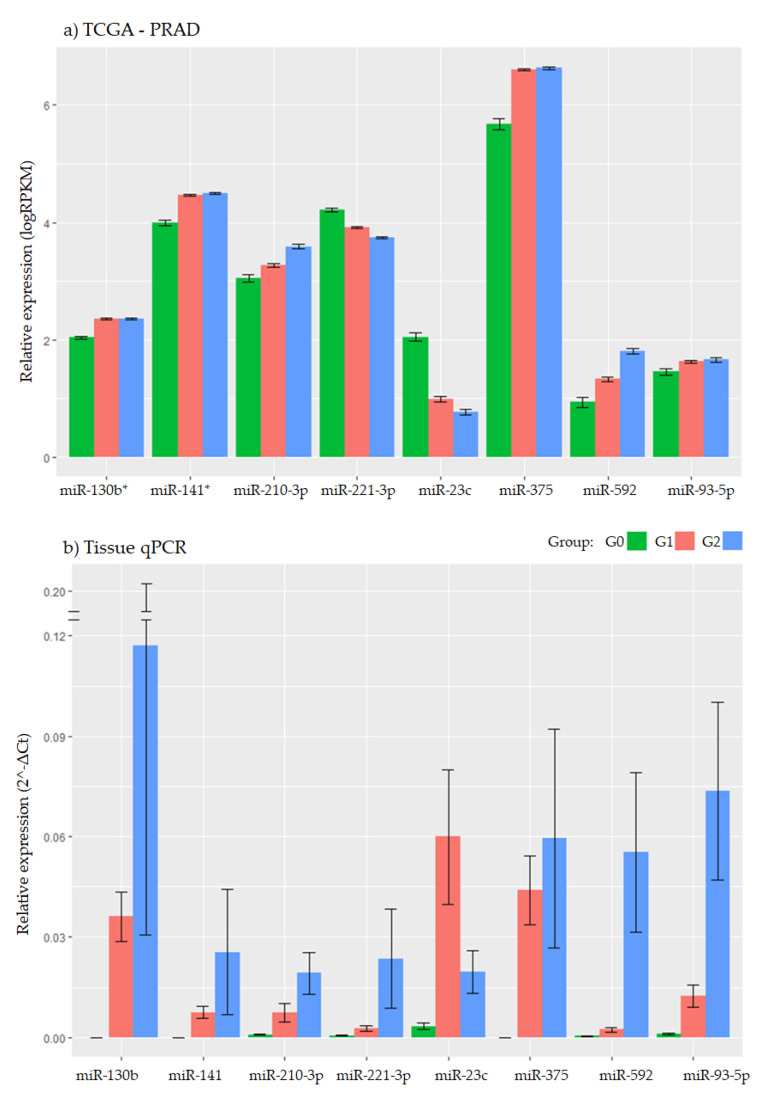
Bar chart of differential expression from qPCR versus TCGA data. (**a**) Plot showing differentially expressed miRNAs when comparing in TCGA (RPKM: Reads per Kilobase Million. * The values expressed in miR-141 and 130b include both -3p and -5p, to be able to compare with the qPCR results. (**b**) Plot showing differentially expressed miRNAs when comparing in qPCR analysis.

**Figure 5 biomedicines-09-00646-f005:**
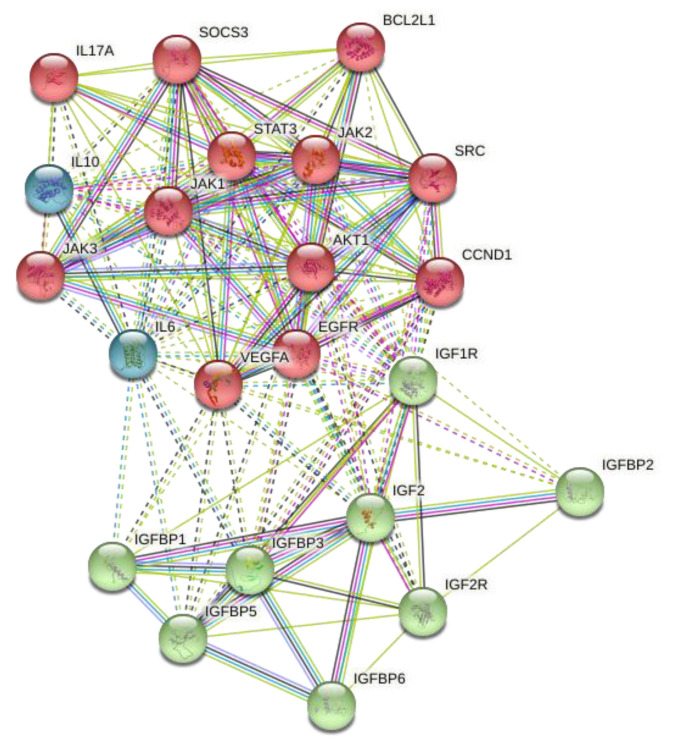
STRING network (default k-means clustering method) performed by the introduction of STAT3 and IGF2 (both modulated by miR-93-5p) with 20 interactors.

**Table 1 biomedicines-09-00646-t001:** Descriptive characteristics of PC tissue samples (*n* = 131).

Variables	Patients
Age average	69.11
Total Age range	48–88
PSA (prostate specific antigen) debut (ng/mL)	
> 4 ≤ 10	72 (55.0%)
> 10 ≤ 20	24 (18.3%)
>20	35 (26.7%)
Gleason Score	
≤7	110 (84%)
>7	21 (16%)
T Stage	
T1–T2	119 (91.54%)
T3–T4	11 (8.46%)
Missing	1
D’Amico risk	
Low	46 (35.38%)
Medium	43 (33.08%)
High	41 (31.54%)
Missing	1
Follow-up (months)	
Mean	32.17
Follow-up range	0–54
Missing	8

**Table 2 biomedicines-09-00646-t002:** Candidate miRNAs for prognosis biomarkers of PC (G1 vs. G2).

miRNA	logFC	F	*p* Value	FDR
**miR-592**	**2.0006**	**150.59191**	**1.14192 × 10** ^ **−30** ^	**6.06360 × 10** ^ **−28** ^
miR-508-3p	−3.41187	135.34909	4.85900 × 10^−28^	1.29006 × 10^−25^
**miR-210-3p**	**1.45741**	**109.56902**	**1.89422 × 10** ^ **−23** ^	**3.35277 × 10** ^ **−21** ^
miR-514a-3p	−2.66782	103.89153	2.06217 × 10^−22^	2.73753 × 10^−20^
miR-509-3p	−2.73042	98.307609	2.20640 × 10^−21^	1.95267 × 10^−19^
miR-708-5p	0.72937	86.798789	3.13431 × 10^−19^	2.37759 × 10^−17^
miR-425-5p	0.58884	71.284172	2.93140 × 10^−16^	1.94571 × 10^−14^
miR-133b	−0.92370	60.11197	4.56314 × 10^−14^	1.86386 × 10^−12^
miR-133a-3p	−0.75474	55.654882	3.52380 × 10^−13^	1.24742 × 10^−11^
miR-217	0.79852	54.514827	5.96134 × 10^−13^	1.97842 × 10^−11^
miR-137	1.78649	51.939089	1.964050 × 10^−12^	5.79394 × 10^−11^
miR-653-5p	0.61648	50.472116	3.884013 × 10^−12^	1.08547 × 10^−10^
miR-221-5p	−1.00961	46.700419	2.263673 × 10^−11^	5.46368 × 10^−10^
miR-7641	−1.33368	44.875201	5.339282 × 10^−11^	1.23267 × 10^−9^
miR-1-3p	−0.59135	43.944679	8.280247 × 10^−11^	1.75872 × 10^−9^
miR-6715b-3p	−1.23979	40.649659	3.944519 × 10^−10^	7.48050× 10^−9^
miR-301a-5p	0.64561	40.571194	4.094489 × 10^−10^	7.49715 × 10^−9^
miR-129-5p	0.92748	38.07787	1.344876 × 10^−9^	2.23165 × 10^−8^
miR-4449	−1.28009	36.42357	2.971930 × 10^−9^	4.64145 × 10^−8^
miR-216a-5p	0.82891	35.55502	4.512166 × 10^−9^	6.84560 × 10^−8^
miR-5680	0.76444	34.34927	8.068259 × 10^−9^	1.19006 × 10^−7^
miR-449a	1.20256	29.54196	8.334028 × 10^−8^	1.02915 × 10^−6^
**miR-23c**	**−0.90493**	**28.19839**	**1.609267 × 10** ^ **−7** ^	**1.81812 × 10** ^ **−6** ^
miR-1298-5p	1.10526	27.73253	2.022880 × 10^−7^	2.23781 × 10^−6^
miR-378d	−0.61904	25.98869	4.775705 × 10^−7^	4.69611 × 10^−6^

Summary reports of miRNAs: down-regulated, 17; up-regulated, 16; total DE miRNAs, 33. Abbreviations: DE (differential expression); logFC (logarithmic fold change); F (quasi-likelihood F-statistic for the GLM (quasi-likelihood F-test)); FDR (False Discovery Rate). Here, comparisons were developed, including tissue samples (480 cases), comparing tumoral area below Gleason ≤7 (G1; *n* = 285) with tumoral tissue area above Gleason >7 (G2; *n* = 195).

## Data Availability

The data that support the findings of this study are available from the corresponding author under reasonable request from L.J.M.-G. and M.J.A.-C.

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
