# Peer review of "Identification of MicroRNAs as Viable Aggressiveness Biomarkers for Prostate Cancer"

_biomedicines, 2021, doi:10.3390/biomedicines9060646_

Round 1

Reviewer 1 Report

Thank your for your detailed revision. 

Author Response

Thank you very much for your review. Thank you again for your time and patient.

Reviewer 2 Report

The conclusion by the Martinez LJ. et al that the miR93-5p is the most promising non-invasive biomarker for PC aggressiveness and diagnosis is very dangerous for the diagnostic field. These statements must be tuned down.

Most information is derived from bioinformatic predictions and literature search.

Assuming in diagnostics a specific expression level of the miR93-5p has been identified in blood of a patient, what exactly is the measure to be a biomarker? Which house-keeping gene in serum will be used? Since these questions were not addressed the title must be changed towards predicted biomarkers. Authors should also not focus on one miR only.

  • Authors must clearly emphasize their arguments of miR93-5p being the klost important one. The listing of other publications about miR93-5p as in a review is not helpful without supporting data within this manuscript.
  • This miR93 does not show up in volcano plot, shows the least regulation in the TCGA data set.
  • The qPCR data would also suggest other miRs being also relevant
  • So, for diagnostics to suggest only one miR, the miR93-5p, is dangerous fort he diangnostic field if no additional data are provided for that specific miR93-5p .

we have performed miRNAs analysis in

change to:

we have performed miRNAs analysis of

Or even targeting FGFRL1 which promotes lung cancer metastasis,

verb is missing

Gleason scores  equal to, or below (G1); and above 7 (G2),

change to:

Gleason scores  equal to, or below 7 (G1); and above 7 (G2),

Tables G0 vs G1 and G0 238 vs G2 in ST4 and ST5.

ST4 and ST5 not defined

G0 versus G1; and G0 versus G2 (see details in SF2 and SF3). SF2 and SF3 not explained

Explain SF1, why not mentioning supplementary Fig. S1, … S2?

3.3. Functional Analysis

 change to:

3.3. Predicted functional Analysis

Author Response

First of all we would like to thanks the reviewer all the changes and suggestions. Thank you again for your time and patient. Just below we will answer point by point to all the comments.

  1. The conclusion by the Martinez LJ. et al that the miR93-5p is the most promising non-invasive biomarker for PC aggressiveness and diagnosis is very dangerous for the diagnostic field. These statements must be tuned down. Most information is derived from bioinformatic predictions and literature search.

Assuming in diagnostics a specific expression level of the miR93-5p has been identified in blood of a patient, what exactly is the measure to be a biomarker? Which house-keeping gene in serum will be used? Since these questions were not addressed the title must be changed towards predicted biomarkers. Authors should also not focus on one miR only.

Thank you for your comments, we will answer in detail just below.

  • It is true that part of the data are obtained form bioinformatic preditions by the analysis in TCGA (The Cancer Genome Atlas) including an extensive cohort (n=531), however we have confirmed top miRNAs in 159 PC fresh tissues and 60 blood samples from prostate cancer patients. We are conscious that the experimental analysis is not developed in an extensive cohort, but we have a follow-up and clinical updated data of all these (tissue and blood) samples. For future analysis, we will enlarge blood samples.
  • At least, we have focused in miR93-5p because after all the analysis (bioinformatic and analytical ones in tissue and blood samples) we think is the best candidate. This affirmation is mainly done because this miRNA is just the one that follows the same patterns in TCGA, qPCR in tissue and blood samples. It is true that according our results is not the only one, and in section “3.2. Molecular analysis” we also highlighted the role of miR-210-3p, miR-23c and miR-592. We know that PC is a very heterogeneous tumor, for that reason according to present data an several previous literature reports we think focusing in this miRNA will help future studies in PC, although we are totally in concordance with the reviewer that just one miRNA will not have the answer.
  • We have also changed the title of the manuscript as you suggested to “Identification of miRNAs as viable Aggressiveness Biomarkers for Prostate Cancer”.
  • For the point of the measure of a biomarker, we have included in the text several techniques that could be used, being cautious that will need a validation process for knowing sensitivity and specificity depending on the sample and the equipment where it will be detected. We here focus on the use of RT-PCR as the easiest way for detecting miRNAs, similarly has been indicated by Cappelletti et al., 2015, Schwarzenbach et al., 2014 and Lan et al., 2015. These authors indicated that “miRNAs circulate in blood and can be easily obtained, and measured through well known, low-cost methods like RT-PCR. Molecules are detectable even in a small sample amount (Cappelletti et al., 2015). Stability in body fluids, easy obtainment and detection are important for their future clinical applications (Schwarzenbach et al., 2014; Lan et al., 2015)”.

Our data are based on RT-PCR using TaqMan technology but it is also possible to use SYBR Green if the probes have enough quality. C. E. Condrat et al. published in Cells 2019 indicated that “the gold-standard for miRNA quantification is quantitative reverse transcriptase PCR (RT-qPCR) with the advantage of having high sensitivity and specificity rates.” Moreover, this author also indicated that “northern blot hybridization represents another widely used alternative for quantitative assessments of miRNAs. This method involves the separation of the total amount of RNA on polyacrylamide gel that possesses the property of denaturation, followed by its transfer on a nylon membrane. However, this technique tends to be strenuous, it necessitates large quantities of RNA and it has sometimes been reported to omit rare types of miRNA. Two other methods are also in use for the same purpose of identifying the miRNAs. In situ hybridization or ISH is a technique that utilizes radioactive, fluorescent or dioxygenin probes to bind the desired RNA, therefore comparing the expression of miRNAs in various cells.The disadvantages of ISH, however, are still significant and include laborious steps and long processes, with a predisposition towards errors. The second method is next-generation sequencing or NGS. Despite the fact that this is a highly accurate technique that has the ability to detect single miRNAs with the precision of one nucleotide, its high costs lead to a limitation of this technique’s wider accessibility”.

In the case we have an increased dataset of miRNAs, we can also use microRNA arrays, for example, Affymetrix microRNA arrays containing 1,734 human mature miRNA probe sets.

As can be seen, it is true that there are a variety of methods and techniques but the most suitable and easy one is quantitative reverse transcriptase PCR (RT-qPCR), the one we have focused in present article. Moreover, if we finally want to include these miRNAs in clinical practices, we have to take into account using easy and reproducible technologies and RT-qPCR seems to be the best option.

  • According to the house-keeping gene in serum, we have to indicated that as it is mentioned in section 2.4 we have developed different studies for choosing the most suitable house-keepings using RNU6, GAPDH and miR-130b. At the end, we obtained that RNU6B is the best option. We replicate the same in blood samples, and after doing the analysis in all the 10 miRNAs of present work we developed an analysis by global mean normalization. According to all these checks we decided using the same house-keeping (RNU6B). Moreover, other authors such as J.Ree 2015 in analysis of plasma miRNAs and study for suitable house-keepings stated that “the use of both RNU6 and 520d-5p as house-keepings provided reliable results”. Moreover, C.H.Hsieh et al.2012 suggested that “Genes typically used for reference in blood are RNU6B and 5S ribosomal RNA. Both suggested as stable and consistent house-keepings for the analysis of expression of whole blood–derived.” Furthermore, Y.Yin et al 2012, indicated that “miR-16 and RUN6B house-keepings are one of the normalization strategies for the analysis of circulating miRNAs”:

We have included a sentence with this idea in the Discussion section as follows “However, as it is previously mentioned by several authors, such as H.P. Liu et al., miRNAs are suitable molecules for being biomarkers in PC with sensitivity, specificity and accuracy values next to 99% in extended cohorts. “

And also, we have included footnotes in supplementary material:

Footnote SF1. Cycle threshold (Ct) values of different tested endogenous controls (RNU6, GAPDH, and miR-130b) in tissue samples. After studying different normalization methods using these previous selected endogenous controls and all the ten miRNAs of present work, we concluded that normalization using RNU6B as control is the best option. In the graph it can be seen that RNU6B normalizer shows less variability and earlier Ct values than miR-130b

Authors must clearly emphasize their arguments of miR93-5p being the klost important one. The listing of other publications about miR93-5p as in a review is not helpful without supporting data within this manuscript.

Thank you for your comments, as we have previously mentioned and it is included in ST6 (a caption of this miRNA is included just below. This miRNA is the one that follows the same patterns in TCGA, tissue and blood samples, although it is true that bioinformatic analysis showed other tops one miRNAs, not all comparisons follow the same patterns in all the analysis.

  • This miR93 does not show up in volcano plot, shows the least regulation in the TCGA data set.

miR93-5p shows a great difference when comparing G0vsG1 and G0vsG2, by contrast it is not differentially expressed in G1vsG2 based on our filters (|FC ≥ 1.5| and FDR < 0.05). Nevertheless, it is over-expressed in G2 as shown in Figure 4.Thus, it is proven that in aggressive stages it is over-expressed, what makes interesting clue for a biomarker. As it is not differentially expressed in G1vsG2 it is not shown in Figure 3 (volcano plot) but it is represented in both volcanos of supplementary information (SF2 and SF3).

  • The qPCR data would also suggest other miRs being also relevant

As it is shown in Figure 4 and ST6, we have demonstrated more relevant results in qPCR but as we have previously mentioned they all not have the same tendency in TCGA and qPCR analysis (blood and tissue), such as in miR-23c. In others such as miR-592 that also showed interesting results, it has also obtained low quality amplification reports (values below Ct 30) in most of the samples, for that reason we did not focus on it.

  • So, for diagnostics to suggest only one miR, the miR93-5p, is dangerous fort he diangnostic field if no additional data are provided for that specific miR93-5p.

We totally share this comment, for that reason in point 3.2. Molecular analysis and Figure 4 we have explained several miRNAs. Moreover, we have discussed and focused on 5 of them and included as a conclusion in abstract section “To sum up, miR-210-3p, miR-23c, miR-592 and miR93-5p miRNAs are suggested to be potential biomarkers for PC risk stratification that could be included in non-invasive strategies such as liquid biopsy in precision medicine for PC management.” and conclusion section.

  • we have performed miRNAs analysis in change to: we have performed miRNAs analysis of

We have done this change.

  • Or even targeting FGFRL1 which promotes lung cancer metastasis, verb is missing

We have done this change.

  • Gleason scores equal to, or below (G1); and above 7 (G2), change to: Gleason scores equal to, or below 7 (G1); and above 7 (G2),

We have done this change.

  • Tables G0 vs G1 and G0 238 vs G2 in ST4 and ST5. ST4 and ST5 not defined

These supplementary tables are mentioned and referred in the main text see section 3.1.1. TCGA Data of Prostate Adenocarcinoma (TCGA_PRAD). We have included footnotes in the Supplemenary Information section for clarifying these tables.

  • Explain SF1, why not mentioning supplementary Fig. S1, … S2? G0 versus G1; and G0 versus G2 (see details in SF2 and SF3). SF2 and SF3 not explained

These all Supplementary Figures are mentioned in the text in sections “2.4. Selection of Candidate miRNA Normalizers” for SF1, “3.1.2. Differential Expression (DE) Analyses” for SF2 and SF3. We have included footnotes in supplementary sections for improving the understanding of them.

Footnote SF1. Cycle threshold (Ct) values of different tested endogenous controls (RNU6, GAPDH, and miR-130b) in tissue samples. After studying different normalization methods using these previous selected endogenous controls and all the ten miRNAs of present work, we concluded that normalization using RNU6B as control is the best option. In the graph it can be seen that RNU6B normalizer shows less variability and earlier Ct values than miR-130b

Footnote SF2. Volcano plot represents miRNAs dispersion according to their expression patterns; using logFC (X axis) versus statistical significance FDR (Y axis). In blue, they are represented those miRNAs that are not differentially expressed contrasting with DE analysis; which are represented in red. As nearer to the left side, miRNAs are under-expressed contrasting with those closer to the right side; which are over-expressed. Moreover, those miRNAs which are on the upper part of the Y axis have a stronger statistical significance. Most interesting miRNAs for present work are identified surrounded by a circle.

Footnote SF3. Volcano plot represents miRNAs dispersion according to their expression patterns; using logFC (X axis) versus statistical significance FDR (Y axis). In blue, they are represented those miRNAs that are not differentially expressed contrasting with DE analysis; which are represented in red. As nearer to the left side, miRNAs are under-expressed contrasting with those closer to the right side; which are over-expressed. Moreover, those miRNAs which are on the upper part of the Y axis have a stronger statistical significance. Most interesting miRNAs for present work are identified surrounded by a circle.

We also included a Footnote in SF6.

  • 3.3. Functional Analysis change to: 3.3. Predicted functional Analysis

We have done this change.

Round 2

Reviewer 2 Report

Authors addressed the criticism in a satisfactory manner

This manuscript is a resubmission of an earlier submission. The following is a list of the peer review reports and author responses from that submission.

Round 1

Reviewer 1 Report

The manuscript by Martinez-Gonzalez et al. analyzed miRNAs from prostate cancer patients from the TSGA data comparing the expression between Gleason Score 7 and below 7, above 7, and non-tumor samples.

In addition, the expression of few selected miRNAs were analyzed from collected tissue samples with confirmed prostate cancer. Authors used bioinformatic tools and identified a list of miRNAs up- or downregulated in prostate cancer samples as well as between Gleason score below and above 7 from TCGA data sets. The level of different expression of 8 miRNAs were experimentally analyzed while a fraction confirmed the TCGA data sets.

Major points:

  1. Authors discuss about biofluids both in introduction and discussion. However, authors do show data from tumor samples and not biofluids. Thus, to avoid misleading delete the paragraphs about using biofluids.

  1. Authors focused only on comparing G0 with G2 and G1 with G2. Authors should provide bioinformatic analysis of changes in expression between G0 and G1 set of samples, which is also helpful.

  1. Figure 5. The labelling is missing at the ordinate.

Figure 5: What is 2,5? Is 2.5 meant?

Figure 5: Error bars are missing. It is expected having so many patient samples to show statistics.

Figure 5: Indicate A and B at the top left side.

  1. Figure 5: In A) Authors compare G0vsG2 from TCGA with G1vsG2 from tissues? Is this a correct comparison? Authors must compare similar tumor gradings

  1. Authors identified some miRNAs being candidates for diagnostic. Authors should provide possible cellular pathways that involve the miRNAs, such as miRNA-23c and -93-5p and others to provide some mechanistic insights.

  1. Authors should discuss critically about the Gleason scores below 7. Is Gleason Score 5 considered as prostate cancer?

  1. The manuscript requires extensive English editing. It sometimes very hard to read.
  2. The citation numbers in the text are not according to Biomedicine regulations. e.g. what is “PC6”. ?

Minor points:

  1. Authors should clearly note in each figure legend from which data set, TCGA or their own, the presented data are derived.

10. Authors differentiate between TCGA samples and tissue samples. However, TCGA based data sets are also derived from tissue samples. Thus make the distinction more clear

Reviewer 2 Report

Dear authors, 

Thank you for submitting paper with extensive bioinformatic analysis. 

I have some comments for your paper. 

1) As you commented, there are always big gaps between research and real clinical practice when looking for efficient biomarker for diagnosis and stratification. 

Your results are just mainly focused on the bioinformatic analysis from TCGA, and I think that your results from molecular analysis for validation don't help much to close the gap between research and real clinical practice. 

it is considered too hasty to say that they are promising biomarkers, just because there is a difference in expression between the two groups. 

When talking about the biomarker research, we have to talk about how it will be used and how it will help when it is actually introduced into the real clinical practice, but such content is not mentioned at all. So such content should be added.

2) miRN-23c, 212-3p shows the contrast results between TCGA and validation, so you should mention why these results happen without saying just interesting role. 

Thanks.  

Reviewer 3 Report

The authors investigated the expression of several miRNAs in prostate cancer PC) tissue samples and found a miRNA signature that correlates with aggressive PC. They first performed a bioinformatics analysis of TCGA data and identified candidate diagnostics miRNAs by comparing healthy areas (G0) of PC tissues and areas of tumors with a Gleason score equal to or below 7 (G1) and above 7 (G2). The expression of candidate miRNA biomarkers was further investigated in 131 fresh tissue samples, but this analysis did not validate the selected miRNA as biomarkers for PC. Therefore, additional miRNAs were selected based on the literature and their potential correlation with PC aggressiveness was also evaluated.  Finally, one miRNA, miR-93-5p, was confirmed in both TCGA and fresh tissue samples.

My recommendation is to revise thoroughly the manuscript for the English grammar/language, since it is nearly impossible to understand the study in the present form.  

Other comments:

  • It looks like the 5 miRNAs identified in the study were already correlated with PC screening, aggressiveness and recurrence, as stated in the discussion (line 269). Therefore, it is not clear the novelty in this work.
  • Are the differences in miRNA expression in figure 5 statistical significant?
  • If I understood correctly, miR-93-5p correlates with a Gleason score, but would this miRNA identify aggressive PC without previous classification of the tissue according to the Gleason score? This would be important if the authors think that miR-93-5p detection can be adopted in the clinic for predicting the aggressiveness of PC.